# Object Tracking for a Smart City Using IoT and Edge Computing

**DOI:** 10.3390/s19091987

**Published:** 2019-04-28

**Authors:** Hong Zhang, Zeyu Zhang, Lei Zhang, Yifan Yang, Qiaochu Kang, Daniel Sun

**Affiliations:** 1Image Processing Center, BeiHang University, XueYuan Road No. 37, HaiDian District, Beijing 100083, China; dmrzhang@buaa.edu.cn (H.Z.); zzybeihang@buaa.edu.cn (Z.Z.); leizhang@buaa.edu.cn (L.Z.); stephenyoung@163.com (Y.Y.); 2College of Information and Computer Sciences, University of Massachusetts Amherst, Amherst, MA 01003, USA; qiaochukang@hotmail.com; 3Software and Computational Systems, DATA61, CSIRO E, Level 1, Synergy Building 801, Black Mountain Science and Innovation Park, Clunies Ross Street, Black Mountain, PO Box 1700, Canberra, ACT 2601, Australia; 4School of Computer Science and Engineering, University of New South Wales, Sydney, NSW 2052, Australia

**Keywords:** Internet-of-Things, edge computing, smart city, object tracking, lightweight computing

## Abstract

As the Internet-of-Things (IoT) and edge computing have been major paradigms for distributed data collection, communication, and processing, smart city applications in the real world tend to adopt IoT and edge computing broadly. Today, more and more machine learning algorithms would be deployed into front-end sensors, devices, and edge data centres rather than centralised cloud data centres. However, front-end sensors and devices are usually not so capable as those computing units in huge data centres, and for this sake, in practice, engineers choose to compromise for limited capacity of embedded computing and limited memory, e.g., neural network models being pruned to fit embedded devices. Visual object tracking is one of many important elements of a smart city, and in the IoT and edge computing context, high requirements to computing power and memory space severely prevent massive and accurate tracking. In this paper, we report on our contribution to object tracking on lightweight computing including (1) using limited computing capacity and memory space to realise tracking; (2) proposing a new algorithm region proposal correlation filter fitting for most edge devices. Systematic evaluations show that (1) our techniques can fit most IoT devices; (2) our techniques can keep relatively high accuracy; and (3) the generated model size is much less than others.

## 1. Introduction

Although a massive shift from on-premise software to cloud computing has been witnessed, as a decentralised compromise, edge computing has enhanced cloud processing for time-sensitive applications. Smart city, as a typical large scale system, is gradually adopting IoT and edge computing, and object tracking, among the most important applications in a smart city, have actually used IoT and edge computing for communication and computation infrastructure. Here is an example of object tracking: In order for self-driving cars to become a reality, those cars need to react to external factors in real-time. The moving objects such as cars and pedestrians always bring considerable uncertainty to the self-driving system. If a self-driving car can track the trajectories of moving objects on the road, it will perceive both normal and sudden movements of its surroundings. Once abrupt lane changing is detected, the car must slow down immediately. If a self-driving car is traveling down a road and a pedestrian walks out in front of the car, the car must stop immediately. IoT and edge computing can definitely benefit this scenario for:Low Latency. The computation is implemented directly on the IoT edge devices. There is no network latency of pushing and fetching data. Thus, the system will respond immediately once acquiring sensor information.Using less network bandwidth. For example, Netflix uses local caches distributed on many cities around the globe to reduce the overall network transfer cost and optimise video streaming for much better overall experience.Privacy. In some occasions, data should stay only close to the users, sending minimal amount of data to cloud for improving user experience, like enabling the user to control remotely.Low Cost. Reducing unnecessary centralised computation and communication resource imply low running cost obviously. For example, Netflix uses local caches distributed on many cities around the globe to reduce the overall network transfer cost.

Besides the above example, in a smart parking lot facility, the tracking system can fetch the trajectory of every vehicles and use the information to guide the guests or detect abnormal driving behaviour. Unmanned Aerial Vehicles (UAVs) can follow its owners automatically and cameras can record one’s movement. However, in IoT and edge computing context, high requirement to computing power and memory of traditional computer vision techniques severely prevent massive object tracking. In this paper, we focus on lightweight object tracking that requires less computation, less memory, and less data transmission. Lightweight object tracking needs an elegant algorithm and implementation.

There are two major problems for the existing tracking algorithms. First, it is the tracking accuracy and robustness. Visual object tracking is a challenging task, especially when faced with difficult tracking conditions e.g., occlusion, object deformation and background cluttering. The ability to handle these difficulties directly influences the tracking accuracy and robustness of a tracking algorithm. Recently, the adoption of discriminative learning method in visual tracking field has greatly contributed to the promotion of tracking accuracy and speed. These methods usually train a classifier online in the first frame and then search among candidate patches using the trained classifier in the following frames. In 2010, Bolme et al. [1] adopted the property of circulate matrix for training a Ridge Regression model as the classifier. Thus, the correlation response can be calculated efficiently in the frequency domain and the proposed tracker, as first discriminative correlation filter (DCF) method, enjoys low computation complexity. However, no feature extraction is implemented in [1], thus, the tracking ability is limited by the feature discrimination. Then, Henriques et al. [2,3] introduced Histogram of Gradient (HOG) [4] features into DCF methods and improved its accuracy and robustness. More inspiring improvement to DCF trackers came from works of Danelijan et al. [5,6] and Galoogahi et al. [7,8], that aimed at fading the boundary effects inherited in the circuit matrix assumption of DCF trackers. In 2017, Galoogahi et al. [8] proposed to solve the zero-aliasing MOSSE filter by ADMM method [9]. The tracker, called BACF, achieves relative high performance on benchmarks and runs quite fast. However, BACF still can not overcome the difficult tracking conditions such as occlusion and object deformation. In our approach, we take BACF as the baseline tracker and design concise frameworks to detect tracking status, and thus, enhance the tracking accuracy efficiently.

Next, we illustrate another issue that object tracking systems for edge computing context are faced with. In order to increase the tracking accuracy and robustness, an obvious solution is to adopt more discriminative image features such as deep-learning features. Deep convolutional neural networks(CNNs) have proven to be super effective on several challenging vision tasks [10,11,12], e.g., object detection and face recognition. Compared to the hand-crafted features such as HOG and Color Naming [13], CNN features significantly improve the tracking ability against the target deformation, occlusion and out-of-plane rotation. However, extracting CNN features from deep convolutional network requires expensive computation and CNN features always have high dimensions. These properties ruin the computational efficiency of DCF method. The method such as [14] runs only 5 fps on a quad-core CPU at about 4 GHz. In 2016, Bertinetto et al. [15,16] firstly employed a Siamese network to combine the feature extraction and classification process jointly, called Siamese-fc. As an end-to-end CNN architecture, it achieves relative high performance owning to the one-shot architecture and Graphics Processing Unit (GPU) acceleration. Although Siamese-fc runs around 86 fps on a GPU device, it requires dense parallel computation thus, conducts high power consumption. Besides, CNN methods require huge memory space, usually above 10 Megabytes, to maintain its network model. Hence, in terms of power consumption and memory usage CNN-based tracking algorithms are not suitable for IoT edge applications.

### 1.1. Motivations

Smart city applications tend to adopt IoT and edge computing broadly for distributed data collection, communication and processing. However, front-end sensors and devices are usually not so capable as those computing units in huge data centres, and for this sake, in practice, engineers choose to compromise for limited capacity of embedded computing and limited memory. Visual object tracking as an important element of smart city is a promising application area. The crucial point for designing tracking systems in IoT edge is to keep a balance between tracking accuracy and computation complexity. After carefully review the recent development of object tracking, we notify that although the implementation of DCF-based trackers is quite efficient, tracking accuracy of the existing methods is not good enough for widespread usage. While, CNN-based methods, though more accuracy and robustness, require high computing complexity and large memory space. Hence, we propose a tracking algorithm with region proposals and tracking status detection schemes. Thanks to the proposed frameworks, our method can overcome difficult tracking conditions such as occlusion, rotation and deformation and, thus, dramatically improves the tracking accuracy. Meanwhile, it is suitable for IoT devices with limited computing capacity and memory space.

### 1.2. Contributions

This paper offers the following contributions:(1)We demonstrate a visual object tracking system for smart city applications using IoT and edge computation with limited computing capacity and memory space. It can fit most low-power consumption IoT devices, e.g., Xilinx SoC platforms and Raspberry
Pi ARM devices, hence provide flexible and energy conservation advantages.(2)We introduce a novel region proposal correlation filter algorithm for lightweight computation tracking. Owing to the concise region proposal scheme and feature combination strategy, the algorithm runs around 40 fps on edge devices and requires only 157 KB on-chip memory. Besides, Response Confidence Level (RCL) is proposed to detect tracking status and update tracking model.(3)Our embedded tracking system with proposed algorithm, surpasses existing systems for IoT in tracking accuracy and robustness. It can precisely track the targets even when difficult tracking conditions, e.g., occlusion, deformation or background cluttering, occur.

The remainder of our paper is organized as follows: In Section 2, we briefly summarize the existing tracking systems and their drawbacks. Then, the theory of discriminative correlation filter, an efficient tracking paradigm, is discussed. Next, we introduce related tracking methods based on region proposals. In Section 3, we firstly demonstrate our object tracking architecture using IoT and edge computing. Second, the proposed tracking algorithm is illustrated in detail. After explaining the solution to boundary effect of DCF-based tracker by zero-aliasing filter, we demonstrate our innovative approach of fetching the region proposals from the correlation response. Then, a robust criterion for judging the tracking status is introduced, based on the response’s peak value, PSR and number of proposals. Section 4 is about the experiment and result. The implementation details of our tracking system in IoT context are firstly put forward. Next, we focus on the system tracking performance. We evaluate our algorithm with other related trackers on two benchmarks. The result shows the computation efficiency of our proposed frameworks on DCF-based tracker. Besides, our tracker requires much less memory space while achieves comparable tracking accuracy and robustness. The last section is a summary of the paper and future prospects for object tracking system using IoT and edge computing are forecasted.

## 2. Related Work

In this section, we discuss the applications of tracking system in IoT and edge computing context and the existing tracking algorithms. The pros and cons of the existing tracking systems are illustrated. Then, DCF-based trackers, which are suitable for edge applications, are introduced in Section 2.1. Then in Section 2.2, some region proposal methods aimed at further enhancing the tracking accuracy are discussed.

### 2.1. Object Tracking System in IoT

Researchers at Gartner estimate that there will be 20 billion IoT devices connected to the Internet by 2020 [17]. In order to process huge computation capacity, traditional approaches are to transmit the data generated by the device to a cloud platform. However, there are three major limitations for cloud computing: transmit bandwidth, power consumption and processing latency. An example scenario is the camera surveillance to detect traffic accidents. If the detection algorithm is deployed at the cloud, the bandwidth requirement for HD videos as well as high latency of network transmission will ruin the efficiency and real-time response of the application, respectively. A sample scenario is a security camera that records the video when an accident happens. The application requires wide bandwidth if the whole video is transmitted and immediate response for accident alert. To overcome the limitations of cloud computing, the edge and fog computing has been introduced. In [18], edge datacenter is proposed as the middle layer to optimize the energy consumption and system latency. It proves that for time-sensitive applications, edge computing is an elegant yet efficient solution. Except for the theoretical studies on IoT edge computing, the solutions to large-scale deployment of IoT edge in industries have attracted considerable attention. Amazon has proposed a solution called AWS IoT Greengrass for time-sensitive applications. Meanwhile, Microsoft has also published Azure IoT Edge to extend the cloud service to the edge side. The advantages of these solutions include security and stability of data transmission, versatile toolkits for software development, low cost for large-scale deployment and so on. Although powerful services have been provided by large companies, specific systems and projects on the IoT edge devices still require careful design.

Visual object tracking is such a time-sensitive application and it shows promising prospect in scenarios such as autonomous vehicles, smart city monitor and robotic vision. When designing object tracking algorithms for the IoT edge, it is important to think about the energy efficiency of the system. The critical point is first to understand the main trick and consumption of the tracking algorithm, and then fully utilize the IoT edge resources. In terms of object tracking applications, Shit, R.C et al. [19] introduced sensor localization in IoT infrastructure and the location information is helpful for a coarse search of the target. Nevertheless, tracking accuracy and robustness are always the top consideration of a tracking system. In traditional tracking system the accurate tracking results guarantee the control of front-end cameras to follow the interested targets. In 2019, Luo et al. [20] proposed a novel module called active tracking. The system directly predicts the actions of the camera rather than the tracking results. However, the reinforcement learning of control system still requires high computation complexity and long latency.

Visual object tracking is a low-latency application and it shows promising prospect in scenarios such as autonomous vehicles, smart city monitor and robotic vision. When applying the object tracking algorithm at the egde of IoT network, energy efficiency is essential, especially when the devices are powered by batteries. Another motivation towards energy profiling and enhancement is to reduce carbon emissions. According to a study published by the Centre for Energy Efficient Telecommunications, the cloud was estimated to consume up to 43 TWh in 2015, compared to only 9.2 TWh in 2012, an increase of 46%. This is roughly equivalent to adding 4.9 million cars to the roads [21]. Thus, it is essential to understand the relationship between tracking algorithms and their respective energy consumption to efficiently utilize the IoT device’s power resources. Meanwhile, tracking accuracy is also an essential consideration.

The power consumption of the tracking algorithm mainly depends on the memory access. Under 45 nm CMOS technology, a 32 bit floating point add consumes 0.9 pJ, a 32 bit SRAM cache access takes 5 pJ, while a 32 bit DRAM memory access takes 640 pJ, which is 3 orders of magnitude of an add operation [22]. If the model of the tracking system is larger than 1 megabyte, we need to store the model in DRAM memory thus, it requires more energy to access the tracker model. Existing deep learning algorithms, e.g., Siamese-fc [15] and HCF [14], needs more than 50 megabytes memory space to contain its network models. Although they achieves the state-of-the-art tracking accuracy, the energy efficiency of the entire system blocks its widespread usage in IoT edge devices. While, some traditional tracking algorithms such as KCF [3] and Struck [23], maintain a small model capacity of less than 1 megabyte, however the tracking accuracy of these methods are much lower than deep learning methods, especially when targets are faced with difficult tracking conditions e.g., occlusion, deformation, out-of-plane rotation and background cluttering.

### 2.2. Introduction to the DCF Tracker

Discriminative correlation filter indicates a category of tracking methodology. It was firstly introduced by [1] in 2010. Since then, plenty of methods are proposed for increasing tracking accuracy and robustness. A typical DCF tracker is trained using the image patch x, centered around the target to fit a desired relation response y of Gaussian distribution. The filter h is obtained by minimizing the following objective function in the spatial domain:(1)$$E\left(h\right)={∥\sum_{d=1}^{D}h^{d}⋆x^{d}-y∥}^{\;2}+\lambda \sum_{d=1}^{D}{∥h^{d}∥}^{2}\;.$$
where ⋆ stands for circular correlation operator. $x^{d}\in R^{N}$ and $h^{d}\in R^{N}$ refers to the *d*th channel of vectorized patch and the corresponding filter respectively [24]. *D* is the number of feature channels, *N* is number of pixels in the image patch, and λ is a regularization factor (λ≥0).

By applying Parseval’s theorem to Equation (Equation 1), filter **h** can be obtained in the frequency domain:(2)$$E\left(\hat{h}\right)={∥\sum_{d=1}^{D}{\hat{x}}^{d}⊙{\hat{h}}^{d}-\hat{y}∥}^{2}+\lambda \sum_{d=1}^{D}{∥{\hat{h}}^{d}∥}^{2}\;.$$
where ${\hat{x}}^{d}$, ${\hat{h}}^{d}$ and ${\hat{y}}^{d}$ are the discrete fourier transfer—DFT of $x^{d}$, $h^{d}$ and y respectively. A ⊙ is the element-wise product between two vectors. Then the optimal filter $\hat{h}$ can be solved efficiently in the frequency domain and h is the inverse DFT of $\hat{h}$.

Since there are more than one training samples during tracking process. An concise update scheme for renewing the regression filter $\hat{h}$ is introduced by [25]. The update scheme can be seen as minimizing a weighted squared error,
(3)$$E\left(\hat{h}\right)=\sum_{j=1}^{m}\alpha_{j}{∥\sum_{d=1}^{D}{\hat{x}}_{j}^{d}⊙{\hat{h}}^{d}-{\hat{y}}_{j}∥}^{2}+\lambda \sum_{d=1}^{D}{∥{\hat{h}}^{d}∥}^{2}\;.$$
where sample index *j* is the frame number and $\alpha_{j}$ is the weight for the training sample in *j*th frame. We can achieve the recursive formula of both nominator and denominator to the solution ${\hat{h}}^{d}$ by setting the $\alpha_{j}=\eta {\left(1-\eta \right)}^{\left(m-j\right)}$ for j>1 and $\alpha_{1}={\left(1-\eta \right)}^{\left(m-1\right)}$. η is called learning rate for the model. Then the *d*th channel of ${\hat{h}}^{d}$ is defined as:
(4)$$\begin{array}{c} {\hat{h}}_{j}^{d}=\frac{{\hat{A}}_{j}^{d}}{{\hat{B}}_{j}^{d}+\lambda } \end{array}$$
(5)$$\begin{array}{c} {\hat{A}}_{j}^{d}=\left(1-\eta \right){\hat{A}}_{j-1}^{d}+\eta {\hat{y}}_{j}⊙{\hat{x}}_{j}^{d*} \end{array}$$
(6)$$\begin{array}{c} {\hat{B}}_{j}^{d}=\left(1-\eta \right){\hat{B}}_{j-1}^{d}+\eta \sum_{i=1}^{D}{\hat{x}}_{j}^{i*}{\hat{x}}_{j}^{i} \end{array}$$
where ${\hat{x}}^{*}$ is the complex conjugate of $\hat{x}$. $\hat{x}$ is the diagonal elements of $\hat{X}$, which is also the DFT of x. The nominator ${\hat{A}}^{d}$ and denominator ${\hat{B}}^{d}$ is updated using the predicted target patch. In the next frame, a searching patch z is cropped from the position of last frame, then the predicted location is at where the maximum value of the response map R in Equation (Equation 7) indicates.
(7)$$R=F^{-1}\left(\sum_{d=1}^{D}{\hat{h}}^{d}⊙{\hat{z}}^{d*}\right)\;.$$

Although the DCF-based trackers enjoy the computation efficiency by transforming to the frequency domain, the circular hypothesis of DFT brings boundary effects to the response map which degrades the tracking performance.

### 2.3. Tracking with Region Proposals

Th region proposal method is firstly introduced for the visual detection. It is designed to replace the sliding window approach by fetching potential object locations from the whole frame. Some of the region proposal methods are based on image segmentation and they treat different image segments as object proposals. Selective Search [26] uses a super-pixel segmentation algorithm in color space to find image segments and then merges the segments to obtain proposals. MCG [27], however, segments the image based on the edge cue on multi-scale hierarchy and merges the segments by edge strength. Usually, only the top one thousand proposals are used for the following process, thus, the proposals are finally rank by a combination of multiple cues, such as location, shape, color and edge. While, the shortcoming of segmentation based approach is the high computational complexity. Though number of potential objects are reduced, it is still time-consuming to fetch the credible proposals. Recently, new region proposal methods are suggested for real-time application. EdgeBox [28] and BING [29] are typical methods for quick object proposals. Two approaches both beg the segmentation process and directly score the candidate windows by the property of edges in the window. EdgeBox gives a high score to the windows containing a large number of edges and BING distinguishes the objects and backgrounds by the edge homogeneous observing that edges of objects have heterogeneous property.

As it is mentioned before, visual tracking is time-sensitive. Region proposal methods without segmentation are more suitable for this kind of task. Edgebox Tracking, EBT [30], was presented in 2016. It applies the EdgeBox as a region proposal method and gets the object-like candidates from the whole frame. The proposed candidates are passed to a structure-SVM classifier in order to tell the target from the other cluttering objects. It is a global search strategy for target detection, however, only the edge information is used for classification so that it suffers when motion blur happens. In 2018, a new target-specific object proposal generation method for tracking (TOPGT) [31] was introduced. Different from EBT, TOPGT combines the edge cue and color cue together to generate the object proposals and then ranks the proposals by shape, color as well as size similarity. After then, it trains a CNN network with the proposals and locates the target with the highest network output score. While, although the trained framework of TOPGT is fairly robust against deformation and motion blur the imbalance of positive and negative samples may lead to target drift when occlusion happens. Besides, it requires the CNN-model update during tracking and, thus, suffers from heavy computation complexity and ruins the real-time property.

## 3. Tracking System in IoT and Edge Computing Context

An overview of our system design is shown in Figure 1:

The idea is to use some front-end cameras that provide us with an interface to fetch videos, a local server that connects to those local cameras and processes their images using machine learning and a computer vision algorithm, then sends the processed data or labels to the cloud, so we can monitor and treat the cameras as sensors by knowing the content of the videos. We can use this to track people or vehicles on the street, receive an alert if something alarming happens in a public space, or even train a custom model and allow authorized personnel to walk around a place. There are many application cases for using the object tracking system.

We are going to use some pan-tilt-zoom (PTZ) camera platforms as our front-end cameras. Use a Zynq-7000 development board as the local server (we also try on our laptops), that will run a pre-trained support vector machine (SVM) model for classifying and detecting objects on images. Afterwards, the tracking algorithm starts to track the detected object and analyzes the trajectory of the target. The trajectory is then fed back to the camera platform and used to control the servo system to follow the tracking target. Thus, our system can keep an eye on the target by the tracking system. The classified data is sent to the cloud securely using a Cloud IoT Core and also leveraging the new gateway feature, so the local server can act on behalf of the camera modules for sending data to the cloud. Then the data is processed in an event-based way using Firebase Cloud Functions, that basically stores data on Firebase to make the data available to internet-connected users on a web interface deployed on Firebase Hosting. Also, the local server will provide a local web interface to monitor the cameras. The project architecture contains five parts as shown in Figure 2.

DeviceListener: Responsible for searching the cameras using mDNS, maintaining a list of devices on the local network and emitting events when devices are online or offline.

ObjectTracker: Locate an object and tracks objects using proposed tracking algorithm.

CloudIoTCoreGateway: Handles all the communication and authentication with Cloud IoT Core and also acts as a Cloud IoT Core Gateway.

WebInterface: Provides a web server that serves a web UI and also a real-time engine to sync data with the browser using socket.io.

EdgeServer: Uses all of the above classes, fetching images from the active devices, passing those images through the tracker, then sending data to Cloud IoT Core and providing the data to the local web interface.

In order to keep the target in view, the tracking algorithm plays an important role in the overall system. In this paper, we focus on the tracking algorithm fitting for IoT and edge computing context. Our algorithm is based on the BACF tracker, which aims at alleviating the boundary effect of DCF trackers. In Section 3.1, we firstly explain the principle of BACF. Next, Section 3.2 clarifies our region proposal framework based on DCF trackers. Then in Section 3.3 innovative criterion to detect the tracking status is proposed based on HOG and color histogram response. At last, the strategy employed to combine multi-cues in our implementation is put forward in Section 3.4.

### 3.1. Baseline Tracker: BACF

In order to transfer the objective function of Equation (Equation 1) to the frequency domain, an assumption of periodic extension of image patch is required to ensure the applicability of the DFT. Due to the periodic assumption, a shifted image patch will introduce repetitive content of center patch other than background samples at the image boarder. It brings frequency aliasing in the frequency domain, while in the spatial domain, the background samples are covered, which makes the filter lack discriminative power. In the BACF method, zero-aliasing correlation filter is employed to handle the problem.

Background-aware correlation filter(BACF) is to minimize the following objective function:(8)$$E\left(h\right)={∥P^{⊤}h⋆x-y∥}^{2}+\lambda {∥h∥}^{2}\;.$$
where P is a T×N binary matrix which holds the mid *T* elements of feature channel x. A ${}^{⊤}$ indicates the transport operator. $x\in R^{N}$ and $y\in R^{N}$, N is the length of sample x. Different from Equation (Equation 1), the filter $h\in R^{T}$ is of length *T*, where T≪N.

Similar to the traditional DCF tracker, Equation (Equation 8) can be learned in the frequency domain for computational efficiency. However, since the zero-aliasing constraint of h is in the spatial domain we keep the regularization in spatial domain. Then, the frequency domain expression of Equation (Equation 8) is as follows:(9)$$\begin{array}{cc} E(h,\hat{g}) & ={∥\hat{x}⊙\hat{g}-\hat{y}∥}^{2}+\lambda {∥h∥}^{2}\; \end{array}$$
(10)$$\begin{array}{cc} s.t.\;\hat{g} & =\sqrt{N}FP^{⊤}h \end{array}$$
where $\hat{g}$ is an auxiliary variable and F is an orthonormal N×N DFT matrix, such that the DFT of signal a can be expressed as $\hat{a}=\sqrt{N}Fa$.

To solve Equation (Equation 9), Augmented Lagrangian Method(ALM) is employed:(11)$$\begin{array}{cc} L(h,\hat{g},\hat{\zeta })= & {∥\hat{x}⊙\hat{g}-\hat{y}∥}^{2}+{\hat{\zeta }}^{⊤}\left(\hat{g}-\sqrt{N}FP^{⊤}h\right)\\ & +\lambda {∥h∥}^{2}\;+\mu {∥\hat{g}-\sqrt{N}FP^{⊤}h∥}^{2}\;. \end{array}$$
where $\hat{\zeta }$ is a N×1 Lagrangian vector in the fourier domain and another regularization term to penalty the zero-padding filter $\hat{g}$ is introduced, which μ is the penalty factor. Equation (Equation 11) can be solved iteratively by Alternating Direction Method of Multipliers (ADMMs) [9]. For simplify the process, we just give the solutions to two subproblems ${\hat{g}}^{⋆}$ and $h^{⋆}$. For more details, please refer to [7,8].

#### 3.1.1. Subproblem ${\hat{g}}^{⋆}$

(12)$$\begin{array}{cc} {\hat{g}}^{⋆} & =arg\;minL(\hat{g};\hat{h},\hat{\zeta })\\ & =\left({\hat{S}}_{xy}+\mu \hat{h}-\hat{\zeta }\right)\circ^{-1}\left({\hat{S}}_{xx}+\mu 1\right)\;. \end{array}$$
where $\hat{h}=\sqrt{N}FP^{⊤}h$. It is easily obtained by applying DFT after masking h with $P^{⊤}$. ${\hat{S}}_{xy}$ and ${\hat{S}}_{xx}$ are the cross-spectral and auto-spectral of sample x with desired response y respectively.

#### 3.1.2. Subproblem $h^{⋆}$

(13)$$\begin{array}{cc} {\hat{h}}^{⋆} & =arg\;minL(\hat{h;g},l)\\ & ={\left(\mu +\frac{\lambda }{\sqrt{N}}\right)}^{-1}\left(\mu g+l\right)\;. \end{array}$$
where $g=\frac{1}{\sqrt{N}}PF^{⊤}\hat{g}$ and $l=\frac{1}{\sqrt{N}}PF^{⊤}\hat{\zeta }$. They can be efficiently calculated by applying inverse DFT of $\hat{g}$ and $\hat{\zeta }$ and then cropped them by matrix P.

#### 3.1.3. Lagrangian Multiplier Update

(14)$$\begin{array}{cc} {\hat{\zeta }}^{\left(i+1\right)} & \leftarrow {\hat{\zeta }}^{\left(i\right)}+\mu \left({\hat{g}}^{\left(i+1\right)}-{\hat{h}}^{\left(i+1\right)}\right) \end{array}$$(15)$$\begin{array}{cc} \mu^{\left(i+1\right)} & \leftarrow min(\mu_{max},\;\beta \mu^{\left(i\right)})\;.\; \end{array}$$
where ${\hat{g}}^{i+1}$ and ${\hat{h}}^{i+1}$ are the current solutions to the above subproblems at iterations i+1 within iterative ADMM. The penalty factor μ is gradually increase within iterative ADMM and β is a increment parameter.

The final discriminative filter $\hat{g}$ is then solved iteratively using Equations (Equation 12)–(Equation 15). Similar to traditional DCF solution, the predicted target location is then obtained by multiplying filter $\hat{g}$ with the feature patch $\hat{x}$ in frequency domain.
(16)$$R_{hog}=F^{-1}\left(\sum_{d=1}^{D}{\hat{g}}^{d*}⊙{\hat{x}}^{d}\right)\;.$$

### 3.2. Region Proposal Method

In this section, we will discuss our region proposal approach based on correlation filters. Unlike the general region proposal methods, such as Selective Search, EdgeBox and BING, that aim at locating possible areas of all class objects, our approach is designed to find the candidate positions that contain similar objects with the tracking target. Specifically, the proposed method treats the correlation filter as a prior knowledge and utilizes the correlation response to fetch similar objects among the searching area. Experimental results show that a high recall ratio can be achieved with only 20 proposals generated, shows the efficiency of our scheme. Then, feature combination strategy can be implied at the proposed candidates to enhance tracking accuracy. For a single-target tracking task, the initial object in the first frame is known in advance. Thus, a more specific region proposal method can be proposed based on the known target model. Unlike general region proposal methods, such as Selective search, EdgeBox and BING, region proposal methods for tracking enjoy the advantage of utilizing the tracking model as a prior. Therefore, the proposed region proposal approach for tracking is computationally efficient embedding into DCF framework meanwhile it achieves high precision rate with less than 20 number.

As we review the implementation of DCF trackers, a discriminative filter h keeps updating during the tracking process to distinguish the target from background surroundings. Then, the response map $R_{hog}$ in Equation (Equation 7) is calculated as the cross-correlation of the filter h with searching patch z. Generally, DCF trackers set only the location with peak response value as the prediction target. However, we notice through experiments that although the regression objective is set as a Gaussian function, the actual response map usually contains multiple peak values and the target can be located at suboptimal peaks, especially when targets are faced with object deformation, background clustering or fast motion. There are two possible reasons. First, the model drifts always happen during the tracking process and as it accumulates the discriminative power of the correlation filter drops. Second, hand-crafted features are not robust against all kinds of tracking scenarios. For example, the hog feature is fragile against target rotation and deformation; thus, distracters may have comparable responses as deformed targets. According to the observations, the region proposal method based on correlation filters are presented. Generally, in DCF trackers, only the maximum value location of the response map is used as the prediction of the target. However, during experiments, we find that sometimes the target does not locate at the maximum response position especially when faced with difficult tracking status, e.g., occlusion, fast motion and deformation. There are two possible reasons for the phenomena. First, in order to limit the frequent aliasing, a hanning window is applied on the sample x, which suppresses the response at the boarder. Second, accumulating of model drift may decrease the discriminative power of the filter and result in a suboptimal response. Further investigation about the position of the target under such circumstances notices that though it is not the maximum value location, it always be a local maximum of the surrounding areas. Based on that, we present our region proposal approach for DCF trackers.

All local maxima of the response map $R_{hog}$ are firstly extracted, where a local maximum is defined as the location with a response value larger than its eight connected neighbors. Next, keep only the local maxima with response values that are larger than 0.6 of the global peak value. are chosen to be the candidates The selected proposals are calculated as follows:(17)$$P_{props}=P_{l_{max}}\;\times \;I_{Loc}\left\{R_{hog}\left(P\right)\;\geq \;0.6Peak)\right\}\;.$$
where $P_{props}$ is the center locations of proposals and $P_{l_{max}}$ denotes local maximum locations. I is an indicator function of value 1 at locations with response greater than 0.6 Peak, which is the global maximum of response map.

After locating the center positions, the proposals are generated by fetching bounding-boxes of the same size at each position. Getting the center locations, we fetch the bounding-boxes using the same patch size in the DCF framework. Besides, non-maximum suppression (NMS) is then used on the proposed bounding-boxes to eliminate heavy overlapped pairs. Figure 3 shows the object proposals generated by our region proposal method on videos from OTB2015 and VOT2016 datasets. From Figure 3, there are two regular patterns summarized: From the given sequences, two observation of patterns are catched: (1) The number of proposals is closely related to tracking status. Number of proposals increases when the target is faced with difficult scenarios and vice versa; (2) ground-truth bounding-box always lays among the proposed candidate bounding-boxes unless tracker drift occurs.

### 3.3. Tracking Status Detection

As mentioned in Section 1, object tracking systems are faced with several difficult statuses, such as occlusion, deformation and fast motion, during the tracking process. The strategies for handling the difficult situations have major effects on the performance of the tracker. The first step towards a useful strategy for complex scenarios is to correctly predict the tracking status. Guan et al. [32] proposed an event-triggered decision model to predict the tracking status. It combines spatial and temporal loss with an event decision tree. We adopt a concise implementation in view of the region proposals from correlation filter response. Section 3.3 elaborates our innovative criterion to judge the tracking status. The criterion is closely related to our region proposal method in Section 3.2. When demonstrating the experiment on region proposal extraction, we notice that more candidates are proposed as the target meets difficult situations. In view of the connection between the number of proposals and tracking status, we suggest a new criterion which combines peak value, Peak-to-Sidelobe Ratio (PSR) and number of proposals to predict the tracking status. Different from the methods from [1,33], the ratio between the current frame and the average value of the past frames is employed to indicate the tracking status. Since the criterion is based on the response map, it is named Response Confidence Level (RCL). The definition is as follows:
(18)$$\begin{array}{cc} \mathrm{RCL} & =\frac{Peak_{K}}{{\bar{Peak}}_{1:K-1}}+\frac{PSR_{K}}{{\bar{PSR}}_{1:K-1}}+\frac{1}{num\;of\;props} \end{array}$$
(19)$$\begin{array}{cc} {\bar{Peak}}_{1:K-1} & =\left(\sum_{i=1}^{K-1}Peak_{i}\right)\;/\;\left(K-1\right) \end{array}$$
(20)$$\begin{array}{cc} {\bar{PSR}}_{1:K-1} & =\left(\sum_{i=1}^{K-1}PSR_{i}\right)\;/\;\left(K-1\right) \end{array}$$
where *K* indicates the *K*th frames among the sequence. $Peak_{K}$ stands for the maximum value of the response map and $PSR_{K}=\left(Peak_{K}-\mu \right)\;/\;\sigma$, where μ and σ represent the mean and standard derivation of the response map, respectively. ${\bar{Peak}}_{1:K-1}$ is the average peak value from the 1th frame to the (K−1)th frame. So does ${\bar{PSR}}_{1:K-1}$.

The tracking statuses have been predicted by RCL and then a new update scheme different from the original DCF tracker is introduced. The details about the filter update scheme is in Section 4.2. Figure 4 shows the efficacious of RCL on sequence Girl2. From the figure we can notify that the RCL significantly drops when faced with difficult tracking conditions occur, e.g., occlusion, rotation, background cluttering and fast motion.

### 3.4. Jointly Detection Using Multiple Cues

Although potential bounding-boxes have been proposed in the methods in Section 3.2, how to determine the most likelihood among proposals is still unresolved. As discussed in Section 2.3, multiple feature cues can considerably improve the accuracy and robustness of tracking performance. Aiming at designing a tracker for real-time applications, the CNN-based feature is not considered because of the computational complexity. HOG feature, which is the statistic of edge cue, is already employed in BACF architecture. Other features like shape or color are taken into account. In [34,35,36], color cue is already proven to be powerful combined with HOG feature on DCF trackers. We also find that color cue is robust against the object deformation and rotation yet weak against illumination change and background cluttering. These properties are just complementary to edge-based features like HOG. thus, the color histogram feature is employed as the other cue for the joint detection in our approach, similar to the choice in papers [36,37].

To distinguish target pixels *x* from background surrounding pixels, a naive Bayes classifier is trained with the color histogram of searching patch *I*. Given a target region T(i.e., the initial bounding-box or the predicted area of DCF tracker) and its surrounding region $S=T^{∁}$ in *I*, the object likelihood of pixels *x* is obtained by Bayes rule:(21)$$P\left(x\;|\;T,S,b_{x}\right)=\frac{P\left(b_{x}\;|\;x\in T\right)\;P\left(T\right)}{\sum_{\Omega \in \left\{T,\;S\right\}}^{}P\left(b_{x}\;|\;x\in S\right)\;P\left(S\right)}\;.$$
We define a symbol $H_{\Omega }^{I}\left(b\right)$ to denote the bth bin of Histogram *H* calculated from region Ω∈I. And $b_{x}$ stands for the bin where color component of I(x) belongs to. The conditional probability can be directly estimate by the color histogram, $P\left(b_{x}\;|\;x\in T\right)=H_{T}^{I}\left(b_{x}\right)/\left|T\right|$ and $P\left(b_{x}\;|\;x\in S\right)=H_{S}^{I}\left(b_{x}\right)/\left|S\right|$, where |·| calculates the total of pixels in the area. Besides, the prior probabilities are approximate as P(T)=|T|/|T+S| and P(S)=1−P(T) respectively. Then, by simplifying Equation (Equation 21), we get (Equation 22).
(22)$$P\left(x\;|\;T,S,b_{x}\right)=\frac{H_{T}^{I}\left(b_{x}\right)}{H_{T}^{I}\left(b_{x}\right)+H_{S}^{I}\left(b_{x}\right)}\;.$$

The object likelihood map is defined as $L_{I}$, where the likelihood of each pixels in map *I* is obtained by Equation (Equation 22). Then, $L_{I}$ is embedded into the region proposal framework to jointly combine color cue and edge cue for detection the target. The most intuitive approach is to use the value of $L_{I}\left(x\right)$, where *x* is the center location of the proposals. However, different from response map of DCF filter, whose regression target is a Gaussian distribution, the maximum response of the likelihood map is usually not located at the center of the target. Thus, we introduce a new method to measure the color similarity of the proposals by accumulating the value of $L_{I}\left(x\right)$ among $x\in \Omega_{k}$, where $\Omega_{k}$ is the bounding-box area of the proposal *k*. Furthermore, the accumulated value can be quickly calculated by integral of likelihood map $L_{I}$.
(23)$$\begin{array}{cc} A(x) & =\sum_{x^{\prime }\leq x}L_{I}\left(x^{\prime }\right) \end{array}$$
(24)$$\begin{array}{cc} R_{color}\left(k\right) & =A\left(x_{k}^{1}\right)+A\left(x_{k}^{4}\right)-A\left(x_{k}^{2}\right)-A\left(x_{k}^{3}\right) \end{array}$$
where A(x) is the integral response of likelihood map $L_{I}$. $R_{color}\left(k\right)$ is the color histogram response of proposal *k*, where $x_{k}^{1}$, $x_{k}^{2}$, $x_{k}^{3}$, $x_{k}^{4}$ are the top-left, top-right, bottom-right and bottom-left corner coordinates of bounding-box proposal *k*.

Then the combined response of proposals *k* is generated by multiplex the response of HOG feature with the response of color histogram together as follows:(25)$$R\left(k\right)=R_{hog}\left(k\right)\times R_{color}\left(k\right)\;.$$

Figure 5 shows the architecture of the region proposal correlation filter. The potential bounding-boxes are proposed from the response of the correlation filter, generated by the baseline tracker BACF. Then, different feature cues can be extracted from the candidates which dramatically enhances the robustness of proposed tracker meanwhile reduces the computation redundancy. In this paper, we combine the edge cue of HOG response with the color cue of histogram to achieve real-time tracking performance of about 30 fps. The efficiency of the overall framework is shown in Section 4.3. Moreover, this architecture can be flexibly expanded to combine other feature cues, including shape, texture or deep CNN features.

When a new frame comes, one loop of region proposal correlation filter algorithm is summarized in Algorithm 1.

**Algorithm 1** Framework of proposed region proposal correlation filter method.**Input:**  z: searching image patch, m×n×c;  g: zero-aliasing filter, m×n×c;  fg: color histogram of foreground object, hbins;  bg: color histogram of background surrounding, hbins;**Output:**  Pos: Predicted object position, [xy];1:Extract the hog feature of the input image patch *x* and translate it to frequency domain z^=fft2(fhog(z)).2:Calculate the response of feature patch with zero-aliasing filter and find the original position with peak response. Posorig=argmax[xy]resph and resph=ifft2(g^⋆⊙z^).3:Generate the proposals from the response map according to Equation (Equation 17) and get final *k* candidates: Pospropk by non-maximum suppression.4:Compute the Response Confidence Level(RCL) in terms of Equation (Equation 18).5:**if**RCL<para.Threshold**then**6:Calculate the object likelihood of each proposals by color histogram according to Equations (Equation 22)–(Equation 24).7:Combine the response of hog feather with the color histogram according to Equation (Equation 22) and choose the proposal with the highest response. Posadj=argmaxkPospropk.8:Predict the object position by multi-cue detection when confidence level is low. Pos=Posadj;9:Keep the zero-aliasing filter g unchanged.10:**else**11:Predict the object position by original response when confidence level is high. Pos=Posorig;12:Update the zero-aliasing filter according to Equations (Equation 12)–(Equation 15);13:Update the color histogram of the foreground and background respectively.14:**end if**

## 4. Experiments and Results

We extensively elaborate the implementation details of our tracking system using IoT and edge computing. The overall system is demonstrate in Section 3. The core of the system is the EdgeServer. It firstly fetches images from the active cameras and then passes those images through the tracker. After the tracking process, EdgeServer sends only compressed information to Cloud IoT Core and provides the data to the local web interface. We will discuss the implementation details of EdgeServer in Section 4.1.

Another critical design in the system is the tracking algorithm, which keeps a balance between tracking accuracy and energy consumption. We evaluate our proposed algorithm on the widely used benchmarks and metrics. The benchmark result of our tracking algorithm compared with the related methods indicate the effectiveness of our region proposal framework on DCF trackers. At the end of this section, several state-of-art trackers are used for comparisons to show the tracking accuracy as well as the computation and memory efficiency of our algorithm.

A brief view of our tracking performance: Table 1 shows the computation efficiency of our algorithm. With all of the trackers tested on the same platform, our tracker achieves top accuracy performance and meanwhile ranks #3 in running speed. Moreover, Table 2 shows a comparison between our method and state-of-the-art CNN-based trackers. Results represent that our algorithm attains comparable tracking accuracy with a model size of only 157 KB. Low computing consumption together with small memory requirement result in energy conservation of the whole procedure.

### 4.1. Flexible Implementation for IoT devices

We implement and test our EdgeServer on Xilinx SoC platform Zynq-7000, with a dual-core Cortex-A9 ARM at 1 GHz operation and an Artix7 FPGA with 512 KB on-chip memory. FPGA acts as the Devices Listener, it fetches video collected by front-end cameras and crops only the interest patches to ARM processor. Besides, it also stores the template model and current image on the on-chip block RAM. Meanwhile, a Linux operating system is running on the ARM Cortex-A9 processor and it works on the remaining jobs as an EdgeServer, including object tracking, IoT core gateway and the Web interface. The gateway collects data from the cameras, sending the processed data to the cloud, and also has a local web UI that runs by default on port 3000. On the web interface, you can access to see the cameras images and tracking results. To process the data being received though Cloud IoT Core, we use Firebase Cloud Function. Although we implement our EdgeServer on a Xilinx SoC platform, as it is shown in Figure 6, the proposed tracking system in fact can be extended to many edge platforms with ARM or X86 cores, e.g., Raspberry
PI, STM32 series and so on. Since the overall system requires only 157 KB memory and runs in real-time tracking performance on a dual-core CPU at 1 GHz, the system can implement flexibly.

As we have mentioned, it is especially important to understand the relationship between algorithms and their respective energy consumption to efficiently utilize the IoT device’s power resources. Based on the platform of our EdgeServer, we dedicate on designing an energy-conservation tracking algorithm on dual-core processing unit and the implementation details about our algorithm are then discussed in Section 4.1.

#### Details for Tracking Algorithm

Followed by [3], the 31 dimensional HOG feature of cell size 4×4 is employed in our BACF filter. The searching area is set to be 5 times of the target size in order to handle difficult tracking situations, e.g., fast motion and occlusion. The regularization factor, lambda in Equation (Equation 9), is set as 0.01. A scale variation process referred to [38] is implemented, with the number of scales and size step setting to 3 and 1.02 respectively. Moreover, the ADMM optimization parameters follow the setting in [8]. The number of iterations is set to 2 to make a balance between performance and computation complexity. The penalty factor μ, is updated by Equation (Equation 15), where β=10 and $\mu_{max}=10^{3}$. Color histogram of bin size 32 is computed for target and background areas. The object histogram is particularly calculated among an area with 0.1 inner-padding of the original target size. Furthermore, a normal Gaussian distribution of σ=0.5 is employed to the likelihood map such that center likelihood slightly stands out.

The discriminative filter for regression as well as the color histogram of the target and background surroundings are both updated in order to catch the target deformation during tracking. However, unlike the traditional strategy, that updates the model per frames, we propose a new update scheme based on the proposed RCL criterion. Only the confidential tracking results are considered as reliable training samples.As mentioned in Section 3.3, RCL is a combination of peak value, Peak-to-Sidelobe ratio and the number of proposals. Through experiment, we observe that when $Peak/\bar{Peak}$ and $PSR/\bar{PSR}$ drop to around 0.8 or there are more than 3 number of proposals, the tracking results are usually unreliable. It usually happens on account of difficult tracking status, e.g., deformation, motion blur or occlusion. Thus, we set the threshold of RCL to be 1.93≈(0.8+0.8+1/3). Tracking results with larger RCL will be used to update the BACF filter and the color histograms. Based on the selective update scheme, the learning rates for BACF filter and color histograms are then set to 0.013 and 0.04 respectively.

### 4.2. Benchmarks and Metrics

Our method is firstly evaluated on the OTB2015 [39] benchmark, which contain 100 sequences. The performance of all trackers is measured by two indicators, namely OverlapPrecision(OP) and DistancePrecision(DP). OP metric measures the percentage of frames with IoU scores that are greater than a threshold, where IoU is the intersection over union(IoU) of predicted and ground truth bounding boxes. We compare the trackers using threshold of 0.5 (IoU > 0.5). The area-under-the-curve (AUC) of plot metric is also employed to rank the trackers. While, DP metric measures the percentage of frames with the center distances of predicted and ground-truth bounding boxes less than a threshold. The percentage at threshold of 20 pixels is reported for all trackers. In the experiment, we apply the One-Pass-Evaluation(OPE) followed methodology in [39].

VOT2016 [40] is also applied to show the robustness of the trackers. VOT2016 dataset contains 60 challenging sequences and it provides a evaluation toolkit to measure the trackers, which will re-initial the tracker to correct position once failure occurs. The expected average overlap is proposed to rank the trackers, which combines the per-frame accuracy and failures.

### 4.3. Accuracy and Efficiency

We evaluate our tracker with 15 state-of-the-art trackers includes KCF [3], Struck [23], fDSST [41], SRDCF [5], MDNet [42], CCOT [6], Siamese-fc [15], HCF [14], TCNN [43], LCT [44], Staple-ca [45], PTAV [46], TOPGT [31], ECO-HC [47], MCCT-H [48] and SAMF [16] on two benchmarks OTB2015 and VOT2016.

The results on the two challenging benchmarks show the accuracy and energy efficiency of our tracker, especially on OTB2015 dataset, our approach surpasses all of the state-of-the-art trackers for real-time application with only 157 KB model parameters.

#### 4.3.1. Comparison to Traditional Tracking Algorithms

We firstly demonstrate the effective of our region proposal framework on DCF tracker by comparing the performance of our method with baseline tracker: BACF [8] and other related trackers: SAMF [38], Staple [36], SRDCFad [49], SRDCF [5] and LMCF [33]. OTB2015 benchmark is used to evaluate the trackers.

Figure 7 shows the results of both DP and OP plots on OTB2015 benchmark. Compared with 6 relevant trackers, our proposed tracker achieves top performance on both DP and OP plot of evaluations, with 84.9% precision and 64.0% overlap respectively. The DP metric gains 3% increments and OP metric gains 2.1% opposed to baseline tracker: BACF. Meanwhile, owing to the concise architecture of combining edge and color cues on region proposals our approach runs at 39.3 fps, even faster compared to the original 35 fps implementation of BACF. The SRDCF tracker, another method to limit the boundary effect of correlation filter, achieves 78.8% precision and 59.8% overlap respectively while only runs less than 10 fps. SAMF and Staple are two related trackers that combine the edge and color cues to jointly track the target. Same feature cues are implemented in our framework, however, on account of the region proposal methodology the combination of edge of color cues becomes more robust and efficient. It can be seen from Figure 4 that compared to SAMF tracker, our approach has 18% gain in AUC scores. With regards to Staple, our tracker also achieves an average of 11% improvement.

Table 1 is a summary of our approach and 6 other relevant trackers. Distance precision indicates the percentage at threshold of 20 pixels error, high DP score means the trackers precisely follow the ground-truth of the target. And then overlap precision measures the percentage with IoU score greater than 0.5, higher OP score guarantees that the trackers predict the exact scale variation of the target. The last column, mean fps, indicates the average running speed among all sequences in the dataset. Since all of the listed algorithm is employed on the same processor, higher fps demonstrates the computation efficiency of the algorithms. Although our tracker is not the fastest algorithm, our tracking performance in DP and OP score both rank #1.

#### 4.3.2. Comparison to CNN-Based Tracking Methods

We compare our algorithm with CNN-based tracker on VOT2016 dataset. When evaluated on VOT2016 dataset, all of the tracking parameters keep exactly the same as its implementation in OTB2015 dataset in order to test the generalization of our proposed tracker. We evaluate the trackers in accordance with accuracy, expected average overlap(EAO) and parameter size. The accuracy measures the average overlap ratio between the predicted bounding box and the ground-truth. And EAO combines the raw values of per-frame accuracies and failures, and then measures the expected no-reset of a tracker run on a a short-term sequence [50]. The model parameter size measures the memory space required by an algorithm. Higher the accuracy or EAO score are, better the tracker is. On the contrary, fewer model parameters indicates less memory access and thus, the algorithm enjoys more energy conservation. The tracking results are shown in Figure 8.

Different from OTB2015 dataset, VOT2016 challenge employs a scheme that tracker will be re-initialized after each drift and count for one failure. Thus, VOT2016 dataset mainly focus on short-term tracking. From Table 2 we notice that all top performance trackers like C-COT, TCNN, MDNet and HCF employ deep-learning feature rather than hand-crafted one because of high distinguish ability of deep feature. Moreover, in terms of our RPCF tracker, it introduces **RCL** to self-detect the possible drift of tracker. And then tracker searches potential objects among region proposals and stops update the target’s models. Even if tracking failure occurs, RPCF tracker may correct itself in the following process. The schemes have less influence in VOT2016 evaluation. Even though our tracker ranks #3 in EAO score, only behind two deep-feature trackers, C-COT and TCNN. However, C-COT and TCNN implement VGG-19 as backbone network to collect deep feature, thus, the model parameter size of the trackers are more than 300 Megabytes. From Table 2 we can notice that comparing with state-of-the-art CNN-based trackers our RPCF tracker has the comparable performance with only 157 KB parameter size. Fewer parameter size illustrates less memory access when processing the algorithm, and thus, leads to low power consumption.

### 4.4. Tracking Performance under Difficult Conditions

Section 4.4 demonstrates the tracking performance of our algorithm under different tracking status. Besides the overall performance of the trackers among all the sequences, there are manually labeled attributes to all of the sequences, which indicate the difficult situations happen during the tracking sequence. The performance on the associated attribute sequences shows the trackers capability of handling the specific tracking situation. There are total of 11 attributes in OTB2015 dataset, namely scale-variation, in-plain rotation, out of view, background cluttering, illumination variation, motion blur, fast motion, deformation, out-of plain rotation, occlusion and low-resolution.

In Figure 9, trackers’ performance under different tracking conditions is shown. Owing to the efficiency of proposed region proposal and multi-cue combination scheme, our tracker overwhelmingly exceeds other trackers under difficult tracking conditions, such as out-of-plane rotation, occlusion, deformation and motion blur. We can tell from Figure 9 that especially when faced with out-of-plane rotation and deformation scenarios occur, the overlap precision gains at least 5% comparing to other trackers. The reason is that although the hog feature of the object changes dramatically under out-of-plane rotation and deformation, other features like color histogram and texture stay unchanged. The region proposal scheme of our approach can generate several candidates and use color histogram to re-locate the target. Together with the proposed RCL criterion to sense the tracking status, RPCF outperforms other trackers under difficult tracking conditions.

## 5. Conclusions

In the paper, an object tracking system using IoT and edge computation is introduced. The proposed system focuses on the edge network design and utilizes Google Cloud as the Cloud IoT core. Specifically, EdgeServer is implemented on a Xilinx Zynq-7000 platform with dual-core ARM and FPGA Artix-7. The edge functions include capturing the images from front-end cameras and tracking interested objects among image sequences, where the tracking algorithm plays a crucial rule. Thus, we design an algorithm to keep a balance between the tracking accuracy and utility of limited resources on edge devices. It achieves the state-of-the-art tracking accuracy, but meanwhile, it requires low computing capacity and memory space. In general, the proposed algorithm can proceed a 1080P image in less than 25 milliseconds on Zynq-7000 platform and rank #1 in accuracy among traditional tracking methods. Besides, our method acquires comparable tracking precision compared to the CNN-based trackers of model size beyond 300 Megabytes, while our parameter size is only 157 kilobytes. It is three orders of magnitude less than CNN-based models. All in all, the proposed accurate tracking system enjoys the lightweight computation and requires small memory space therefore it has a promising prospect in IoT and edge computing applications. The algorithm is flexibly deployed to many energy-efficient edge devices.

The proposed tracking algorithm plays as the core of our object tracking system. It maintains an ingenious balance between the tracking performance, system latency and energy consumption. Besides, the C++ implementation of the overall algorithm has the advantage of easy and rapid deployment. The system is easily rearranged from Google Cloud to other IoT platforms such as Microsoft Azure IoT or Amazon AWS Greengrass software merely by designing new interface programs. In the smart city applications, plenty of front-end cameras have been deployed for surveillance, Human–Computer Interaction and sports broadcasting. Our tracking system will improve the existing systems by introducing tracking trajectory of the interested characters. In the future, self-driving cars with the object tracking system will probably sense the movement of surrounding cars and respond instantly to unusual moves.

Although the proposed tracking system achieves the state-of-the-art, we argue that there is still limitation existing in our tracking system. Specifically, it lies in improving tracking robustness for long-time object tracking. An issue exists that a tracker should keep stable when occlusion and fast motion occur; on the contrary, when faced with deformation and rotation it should update rapidly to adapt the appearance changes. However, a reliable method to predict tracking status and its corresponding updating scheme still needs further study. 

## Figures and Tables

**Figure 1 sensors-19-01987-f001:**
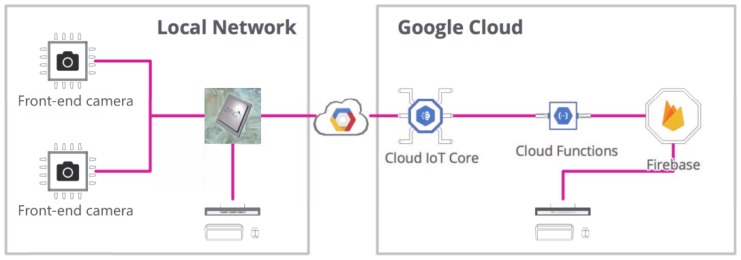
The system design of our tracking system in IoT and edge computing context.

**Figure 2 sensors-19-01987-f002:**
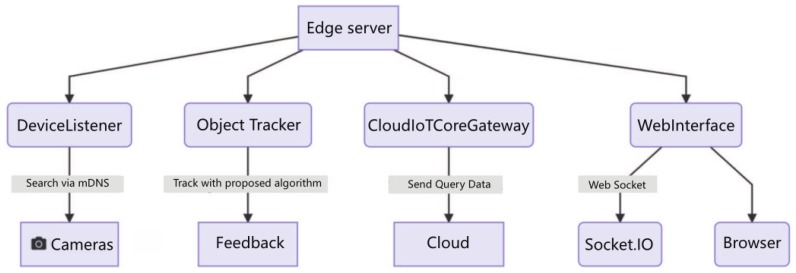
The project architecture of the overall system.

**Figure 3 sensors-19-01987-f003:**
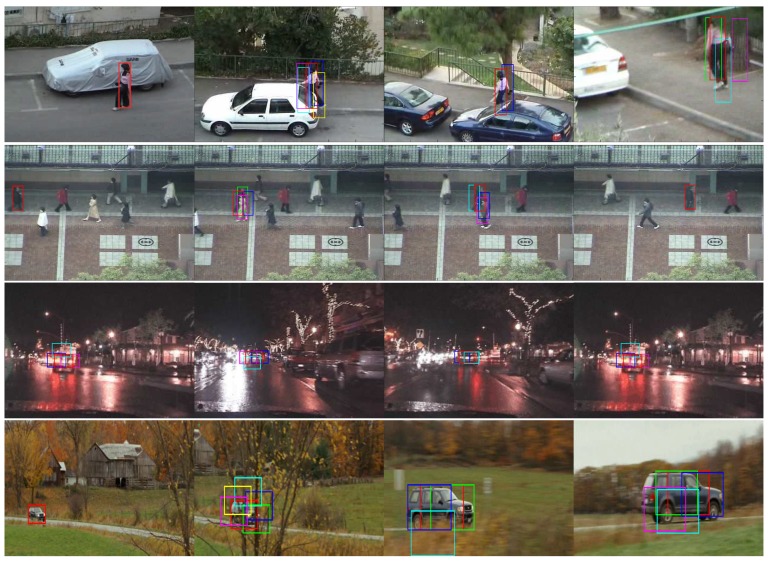
Illustration of the proposed DCF-based region proposal method on four different sequences.

**Figure 4 sensors-19-01987-f004:**
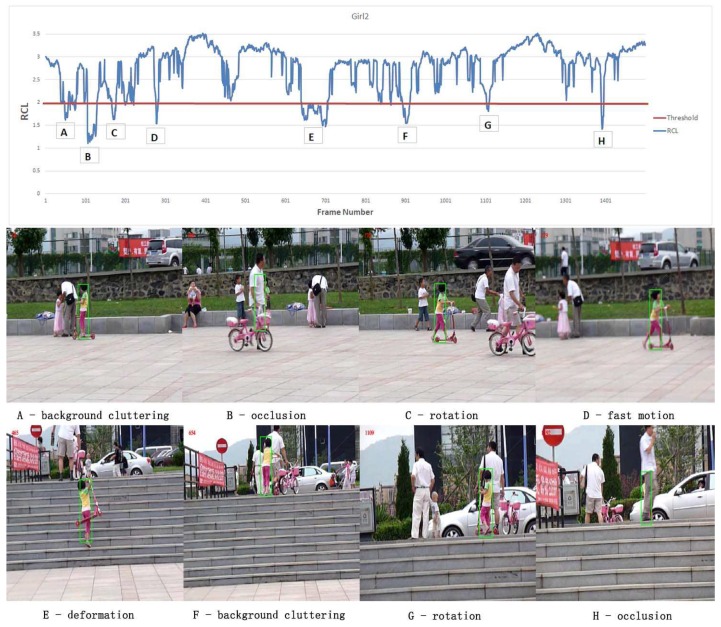
The correspondence between difficult tracking conditions and our proposed **RCL** criterion on test sequences girl2.

**Figure 5 sensors-19-01987-f005:**
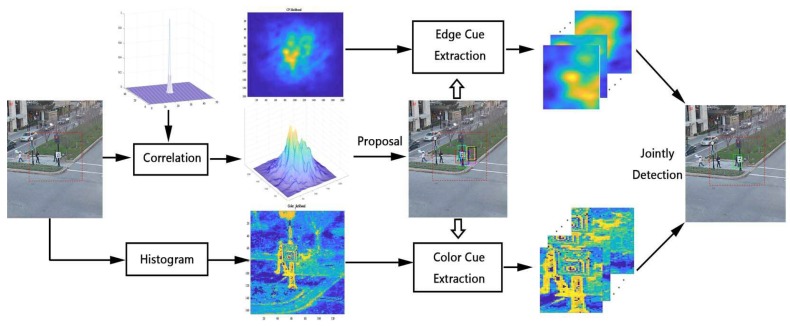
Architecture of the overall Region proposal correlation filter.

**Figure 6 sensors-19-01987-f006:**
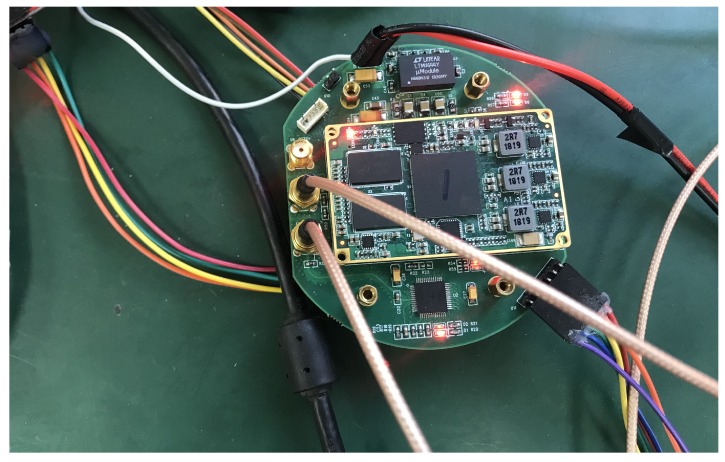
Zynq-7000 platform as the EdgeServer.

**Figure 7 sensors-19-01987-f007:**
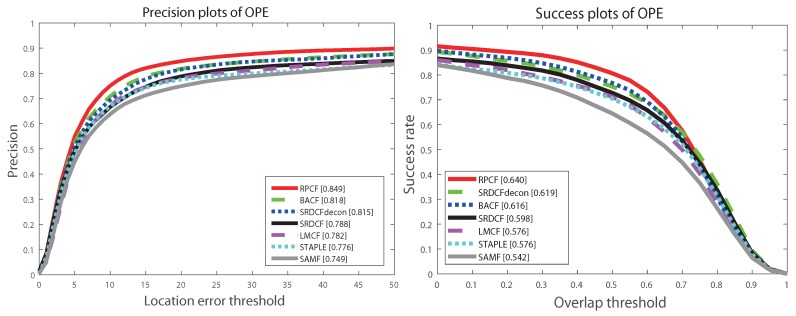
OTB2015 results of related trackers.

**Figure 8 sensors-19-01987-f008:**
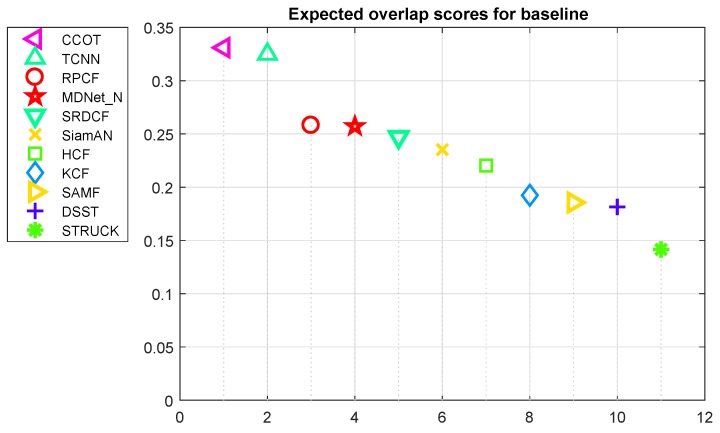
Expected overlap scores in VOT2016 challenge for state-of-the-art trackers.

**Figure 9 sensors-19-01987-f009:**
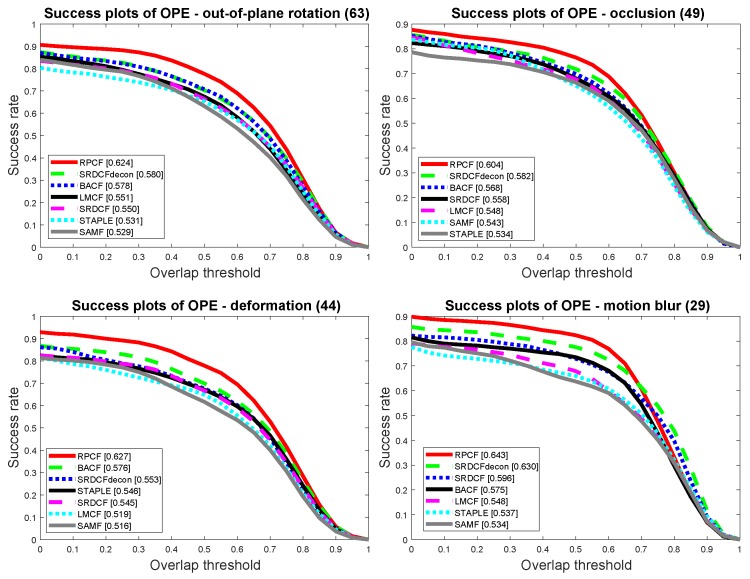
Different tracking conditions analysis of OTB2015.

**Table 1 sensors-19-01987-t001:** Detail information about related trackers performance in OTB100. **Boldface**, *italics* and underline represent 1st, 2nd and 3rd respectively.

Trackers	Comparison	Distance Precision	Overlap Precision	Mean fps
BACF	Boundary Effects	*81.8%*	*77.5%*	35.3
SRDCF	Boundary Effects	78.8%	73.5%	5.6
SAMF	Feature Combination	74.9%	64.9%	18.3
Staple	Feature Combination	77.6%	71.1%	*67.2*
SRDCFad	Tracking Status Prediction	81.5%	76.1%	2.9
LMCF	Tracking Status Prediction	78.2%	71.8%	**85.2**
RPCF	our approach	**84.9%**	**81.6%**	39.3

**Table 2 sensors-19-01987-t002:** Summary of state-of-the-art trackers’ performance on VOT2016. **Boldface**, *italics* and underline represent 1st, 2nd and 3rd respectively.

Tracker	EAO	Accuracy	Parameter Size	
SiamAN	0.235	0.539	14 MB
STRUCK	0.142	0.422	2 MB
SRDCF	0.247	0.532	*1.4 MB*
CCOT	**0.331**	0.535	329 MB
MDNet	0.257	*0.543*	31.6 MB
HCF	0.220	0.435	510 MB
TCNN	*0.325*	**0.553**	491 MB
RPCF	0.260	0.540	**157 KB**

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
