# Peer review of "Object Tracking for a Smart City Using IoT and Edge Computing"

_sensors, 2019, doi:10.3390/s19091987_

Round 1

Reviewer 1 Report

This paper proposed on object tracking on lightweight computing including 1) using limited computing capacity and memory space to realise tracking, 2) proposing a new algorithm region proposal correlation filter fitting for most edge devices. I am happy with the overall contribution, but author should address following comments before final acceptance.

How does background-aware correlation filter works? It needs to be improvised with edge concept.

Why did correlation filters use for region proposal method? It should show clearly reflect in Section 3.2.

Experiment setup could be improvised for better understanding.

Authors could improvise the state-of-the-art by citing recent works as follows.

·         DOI: 10.1109/COMST.2018.2798591

·         DOI: 10.1109/TPAMI.2018.2818132

·         DOI: 10.1109/TPAMI.2019.2899570

·         DOI: 10.1109/ACCESS.2018.2872722

·         DOI: 10.1109/TIE.2018.2835390

·         DOI: 10.1109/TPDS.2017.2787747

Author Response

Thank you for your comments. My reponse can be found in the attachment and a new version of manuscript is uploaded. 

Reviewer 2 Report

This is a good paper on a very interesting area of visual object tracking in the context of smart cities. Focus of the work is on lightweight computing with authors proposing  an algorithm for region proposal correlation filter.

This work has good technical, theoretical and implementation depth with these aspects well explained.  The evaluation appears quite thorough including with a well-known benchmark, and a number of known trackers. 

Two aspects that could be further improved: conclusion section is quite short and I wonder there is a plenty that the community can learn from this use case, implementation, and the evaluation. I would ask the students to expand on conclusions while thinking about the reusability of your work, and its applicability in other contexts/cases. Second, there are a number of grammatical issues throughout the paper that needs a careful consideration. 

Author Response

Thank you for your comments. My reponse is in the Word file and a new version of manuscript is uploaded. 

Reviewer 3 Report

In this paper, the authors demonstrate a visual object tracking system for smart city applications using devices with limited computational capability and memory. Secondly, the authors introduce a novel region proposal correlation filter algorithm for lightweight computation tracking. The following are suggestions. The authors may consider revising the paper accordingly. - "object tracking" term is not clear to me as it is used by various domains differently. the authors may consider defining this term with an example. the car example should be elaborated more clearly. - The abstract may be revised. The motivation and unique challenges related to research are not clear to me. - The authors may consider to include recent technologies progress in industries. For instance, Amazon has released Greengrass technologies to address some of the challenges mentioned by this paper, Microsoft has released IoT Edge. These works should be mentioned in the related work section. - Why do we need to devise a new method/technologies when existing technologies are addressing the challenges? - What is the scope of the proposed method? What are the limitations of the proposed approach?

Author Response

Thank you for your comments. I have uploaded my response  as a Word file and a new version of manuscript is uploaded. 
